# Synthesis of Novel Bromophenol with Diaryl Methanes—Determination of Their Inhibition Effects on Carbonic Anhydrase and Acetylcholinesterase

**DOI:** 10.3390/molecules27217426

**Published:** 2022-11-01

**Authors:** Necla Oztaskin, Suleyman Goksu, Yeliz Demir, Ahmet Maras, İlhami Gulcin

**Affiliations:** 1Department of Chemistry, Faculty of Science, Ataturk University, Erzurum 25240, Turkey; 2Department of Pharmacy Services, Gole Nihat Delibalta Vocational High School, Ardahan University, Ardahan 75000, Turkey

**Keywords:** bromophenol, diaryl methane, demethylation, carbonic anhydrase, acetylcholinesterase

## Abstract

In this work, nine new bromophenol derivatives were designed and synthesized. The alkylation reactions of (2-bromo-4,5-dimethoxyphenyl)methanol (**7**) with substituted benzenes **8**–**12** produced new diaryl methanes **13**–**17**. Targeted bromophenol derivatives **18**–**21** were synthesized via the O-Me demethylation of diaryl methanes with BBr3. Moreover, the synthesized bromophenol compounds were tested with some metabolic enzymes such as acetylcholinesterase (AChE), carbonic anhydrase I (CA I), and II (CA II) isoenzymes. The novel synthesized bromophenol compounds showed Ki values that ranged from 2.53 ± 0.25 to 25.67 ± 4.58 nM against hCA I, from 1.63 ± 0.11 to 15.05 ± 1.07 nM against hCA II, and from 6.54 ± 1.03 to 24.86 ± 5.30 nM against AChE. The studied compounds in this work exhibited effective hCA isoenzyme and AChE enzyme inhibition effects. The results show that they can be used for the treatment of glaucoma, epilepsy, Parkinson’s as well as Alzheimer’s disease (AD) after some imperative pharmacological studies that would reveal their drug potential.

## 1. Introduction

Nature is an important source in drug development research [1]. Marine life is one of the sources that produce naturally occurring bromophenols. In the last decades, there have been many studies on the isolation of bromophenols from marine algae [2,3,4], sponges [5,6], ascidians [7], and corals [8]. In these studies, all these natural bromophenols showed important biological activities. For instance, 5,5′-methylenebis(3,4-dibromobenzene-1,2-diol) (**1**), isolated from the marine algae *Rhodomela confervoides* and *Leathesia nana* showed anti-cancer activity [9]. In another research work, the isolation of 3,4-dibromo-5-(2-bromo-6-(ethoxymethyl)-3,4-dihydroxybenzyl)benzene-1,2-diol (**2**) from red alga (*R. confervoides*) and its antidiabetic activity were reported [10]. Naturally occurring 3,4,6-tribromo-5-(2,5-dibromo-3,4-dihydroxybenzyl)benzene-1,2-diol (**3**), derived from the red alga *Symphyocladia latiuscula*, has been proven to inhibit the aldose reductase enzyme [11]. The isolation from the red algae *V. lanosa* of 3,4-dibromo-5-(2-bromo-3,4-dihydroxy-6-(hydroxymethyl)benzyl)benzene-1,2-diol (**4**) together with glucose 6-phosphate dehydrogenase and their antioxidant properties has also been addressed [12]. In addition, compound **1** has been reported to have isocitrate lyase [13], cytotoxicity [14], antimicrobial [15], and feeding deterrent [16] properties. Moreover, it has been reported that compounds **2** and **4** have antibacterial activities [17] (Figure 1).

In our ongoing project on the total synthesis and biological evaluation of natural bromophenols and their derivatives, we have already reported the first synthesis of bromophenols **1** [18], **2** [19], and **3** [20]. In these studies, the antioxidant properties of **1**, the CA inhibition effects of **2** and **3** were described [18,19,20]. From our early studies, we concluded that not only naturally occurring bromophenols but also their synthetic derivatives, including 4-(2-bromo-4-hydroxybenzyl)benzene-1,2-diol (**5**) and 2-benzyl-5-bromobenzene-1,4-diol (**6**), exhibit CA, AChE, and BChE inhibitory properties [21,22,23] (Figure 1).

Carbonic anhydrases (CAs) catalyze the reversible hydration of water and carbon dioxide (CO_2_) to protons (H^+^) and bicarbonate ions (HCO_3_^−^) [24,25,26]. They take part in a variety of physiological functions, such as ion transport, fatty acid metabolism, bone resorption, pH regulation, and gas exchange. Furthermore, edema and glaucoma occur when the activity of CAs reaches abnormal levels [27,28,29]. Sulfonamides are used as CA inhibitors [30], including N-substituted phthalazine sulfonamides [31], sulphonamide Schiff bases [32], imidazolinone-based benzenesulfonamides and thiourea-substituted benzenesulfonamides [33], imidazolinone-based benzenesulfonamides [34], pyrazoline benzensulfonamides [35,36,37], hetaryl sulfonamides [38], phenolic sulfonamides [39], and quinazolin-sulfonamide [40]. However, various sulfonamides unspecifically block all CA isoforms, which results in adverse side effects. The development of non-sulphonamide-based CAIs is necessary because a sizable section of the population cannot be treated with sulphonamides due to sulfa allergies [41].

By hydrolyzing the neurotransmitter acetylcholine (ACh), the enzyme acetylcholinesterase (AChE) modulates cholinergic transmission at the synaptic level [42,43]. AChE affects cell adhesion, proliferation, and differentiation; the formation of tumors, apoptosis, and amyloid protein deposition in organs as well as AChE are all important cholinergic functions [44,45,46]. Abnormal levels of AChE are associated widely with neurodegenerative disorders such as myasthenia gravis, Parkinson’s disease (PD), and Alzheimer’s disease (AD). Currently, oral active AChE inhibitors that only provide palliative, symptomatic relief are the mainstay of treatment for AD [47,48,49].

The construction or extension of chemical libraries is very important for the development of novel lead compounds in the field of drug design and discovery. Therefore, in this study, we synthesized some novel bromophenols and evaluated their hCA I, hCA II, and AChE inhibitory properties.

## 2. Results and Discussion

### 2.1. Chemistry

In this study, novel bromophenol derivatives **18**–**21** were synthesized in two steps. To synthesize desired diaryl methane compounds **13**–**17**, compound **7** was first synthesized according to the procedure described by Crombie and Josephs [50]. The alkylation of substituted benzenes is a very important reaction for the synthesis of novel alkyl benzenes. The synthesis of diaryl methanes can be achieved via the reaction of benzylalcohol with substituted benzenes in the presence of AlCl_3_ [51]. The application of this methodology to (2-bromo-4,5-dimethoxyphenyl)methanol (**7**) and benzene derivatives **8**–**12** in CH_2_Cl_2_ (DCM) in the presence of AlCl_3_ afforded novel compounds **13**–**16** and a known compound **17** [52], with good yields (75–92%). The O-Me demethylation of arylmethyl ethers with BBr_3_ is an important strategy for the synthesis of bioactive phenols [21]. Therefore, the targeted novel bromophenols **18**–**21** were synthesized from the demethylation reaction of **13**–**16** with BBr_3_ in DCM, with the yields ranging from 73 to 82% (Figure 1). The structures of all the compounds described in this paper were characterized by IR, elemental analysis, and the ^1^H and ^13^C-NMR techniques.

### 2.2. Biochemistry

Since abnormal levels or behaviors of the majority of the sixteen hCA isoenzymes have frequently been linked to several human diseases [53,54,55]. These CA isoforms are intensively found in different tissues and are involved in many important mechanisms such as electrolyte secretion, cell differentiation, bone resorption, calcification, pH and CO_2_ homeostasis, gluconeogenesis, and neurotransmission in mammals [56,57,58]. Hence, many pharmaceutical uses have notable goals for a variety of CA isoforms, including antiglaucoma drugs, anticonvulsant factors/diagnostic, diuretics, antiobesity, and antitumor tools [59,60]. For instance, inhibitors of the hCAs IX and XII isozymes have been used as antitumor and antimetastatic agents [61,62].

High amounts of the hCA I isoform have been found in the red blood cells and the gastrointestinal tract of mammals. The inhibition of this enzyme can be a key component in the treatment of conditions or diseases, including cerebral and retinal edema [63,64]. The enzyme results are given in Table 1 and Figure 2. In the current study, all the novel, synthesized a series of bromophenols (**13**–**21**) efficiently inhibited the hCA I isozyme, with IC_50_ values ranging from 12.38 to 38.50 nM and K_i_ values ranging from 2.53 ± 0.25 to 25.67 ± 4.58 nM. The compound 1-bromo-4,5-dimethoxy-2-(5-methoxy-2-methylbenzyl)benzene (**14**) demonstrated the best inhibition (K_i_: 2.53 ± 0.25 nM) (Figure 2). However, the K_i_ values of novel compounds (**13**–**21**) towards hCA I were decreased as follows: **14** (2.53 ± 0.25 nM) > **15** (9.35 ± 1.88 nM) > **21** (11.00 ± 3.83 nM) > **18** (12.49 ± 0.66 nM) > **16** (12.80 ± 0.52 nM) > **20** (13.37 ± 2.29 nM) > **17** (18.76 ± 4.97 nM) > **19** (20.35 ± 2.92 nM) > **13** (25.67 ± 4.58 nM). Similarly, all the novel, synthesized a series of bromophenols (**13**–**21**) demonstrated competitive inhibition against hCA I isozyme. According to Table 1, the binding of the bromo group to the 4th position of compound **16** caused a 1.37-fold decrease in the K_i_ value, which increased the inhibition efficiency (**15**, K_i_: 9.35 ± 1.88 nM). In compound **16**, methyl bonding (**14**) instead of the methoxy group showed a 5.06-fold greater effect on inhibition. The methoxy group instead of the -OH group in the compounds were more effective in inhibiting hCA I. For example, when compounds **14** and **19** are compared with each other, there is an 8.04-fold difference in the inhibition value.

The dominant cytosolic hCA II isoform plays a critical function in disorders such as glaucoma [65]. In fact, the production of HCO_3_^−^ acts as a method to introduce water and Na^+^ ions into the eye, increasing intraocular pressure. As a result, hCA II isozyme inhibition lowers HCO_3_^−^ generation and eye pressure [66,67]. In the current study, bromophenols (**13**–**21**) effectively inhibited hCA II with IC_50_s ranging from 7.45 to 27.72 nM and K_i_s ranging from 1.63 ± 0.11 to 15.05 ± 1.07 nM. Compound **13** demonstrated the best inhibition effects (K_i_: 1.63 ± 0.11 nM) (Figure 2). When the K_i_ values of the studied compounds (**13**–**21**) were evaluated against hCA II, the following order was found: **13** (1.63 ± 0.11 nM) > **15** (2.62 ± 0.13 nM) > **14** (4.28 ± 0.86 nM) > **21** (4.97 ± 0.59 nM) > **20** (6.21 ± 1.01 nM) > **16** (7.77 ± 0.57 nM) > **18** (9.15 ± 1.36 nM) > **17** (10.33 ± 1.88 nM) > **19** (15.05 ± 1.07 nM). Furthermore, all the novel, synthesized a series of bromophenols (**13**–**21**) exhibited competitive inhibition against the physiologically dominant hCA II isoenzyme. The proposed interaction between the most powerful bromophenols (**20**) and the CA II isoforms is illustrated in Figure 3. Bromophenol (**20**) has two dihydroxy benzyl rings. A second hydrogen bond was modeled between the oxygen atom, which attached to the -OH group at the phenol moiety in the *ortho*-position, and the amide NH of Thr199, a universally conserved amino acid residue in CAs. Thus, phenolic compounds and derivatives bind non-classically to CA, providing clues for the identification of new types of CA inhibitors. Such inhibition mechanisms of phenolic compounds, including bromophenols, are known [68,69]. As shown in Table 1, the attachment of three methoxy groups caused a decrease in the hCA II inhibition value. The methoxy group at positions 2, 3, and 4 may have created a steric hindrance in enzyme inhibition. As in the hCA I isoform, the presence of the methoxy group instead of the hydroxyl group in the compounds was more effective in inhibiting hCA II. When the compounds of **19** and **21** were compared with each other, the presence of the hydroxyl group instead of the methyl group caused a 3.03-fold increase in the inhibition value. A similar situation was observed in hCA I inhibition. This may be because the hydroxyl group is more electronegative than the methyl group.

ACh is used as a neurotransmitter component, and AChE is a crucial enzyme that catalyzes ACh breakdown. This enzyme has been linked to therapeutic targets for AD [70,71]. The hypothesis was put forth to explain AD that synaptic depression is hampered because the cholinergic neuron cells impede ACh hydrolysis [72,73]. ACh hydrolysis is hindered because of AChE inhibition. As a result, the development of AChE enzyme inhibitor drugs and/or modulators is of great interest because it is currently one of the main goals in the fight against AD [74,75]. In the current study, bromophenols (**13**–**21**) effectively inhibited AChE with IC_50_s ranging from 8.35 to 21.00 nM and K_i_s ranging from 6.54 ± 1.03 to 24.86 ± 5.30 nM. The inhibitor effects of the studied compounds (**13**–**21**) against AChE were decreased as follows: **21** (6.54 ± 1.03 nM) > **18** (7.92 ± 1.38 nM) > **20** (8.32 ± 0.69 nM) > **13** (11.04 ± 0.61 nM) > **14** (11.62 ± 2.75 nM) > **16** (16.27 ± 2.98 nM) > **19** (17.43 ± 3.15 nM) > **17** (21.04 ± 4.72 nM) > **15** (24.86 ± 5.30 nM). In addition, all the novel synthesized a series of bromophenols (**13**–**21**) showed competitive inhibition against the cholinergic enzyme of AChE. As shown in Table 1, in the methoxy-bonded compound groups, the fact that the methyl group (**14**) is attached instead of the bromine ion (**13**) did not cause any change in inhibition. When compounds **15** and **16** are compared, the addition of the bromine group to the 4th position caused a rise in the inhibition value. The presence of the -OCH_3_ groups in the middle position without the bromine group was more effective in AChE inhibition (**15**, K_i_: 24.86 ± 5.30 nM; **16**, K_i_: 16.27 ± 2.98 nM). As in hCA I and II, the presence of the methoxy group instead of the hydroxyl group in the compounds was more effective in inhibiting AChE. When compounds **19** and **21** were compared with each other, the presence of the hydroxyl group instead of the methyl group caused a 2.67-fold increase in the inhibition value. 

## 3. Materials and Methods

### 3.1. General

Commercially purchased chemicals were used without further purification. Solvents were used after distillation or after drying with various drying agents. The melting points were determined using a capillary melting equipment and were not corrected (Buechi 530). A PerkinElmer spectrophotometer was used to collect IR spectra (Lancashire, Great Britain) from liquids in 0.1 mm cells. On a 400 (100) MHz (Varian, Danbury, CT) and 400 (100) MHz (Bruker, Fallanden, Switzerland) spectrometers, the ^1^H and ^13^C NMR spectra were collected; d was in ppm, with Me_4_Si as the internal standard. On a Leco CHNS-932 apparatus (St. Joseph, Missouri, USA), elemental analyses were performed. The silica gel was used for column chromatography (60-mesh, Merck, Darmstadt, Germany). PLC stands for preparative thick-layer chromatography, which used 1 mm of silica gel (60 PF, Merck, Darmstadt, Germany) on glass plates. The synthesized compounds’ ^1^HNMR and ^13^C NMR spectra are provided as Appendix A.

### 3.2. Chemistry

The synthesis of compound **7** was performed according to procedure of Crombie and Joseph [50]. The synthesis of compounds **8**–**12** were carried out according to the method given in the literature [51]. The compound 1-(2-bromo-4,5-dimethoxybenzyl)-2,3,4-trimethoxy benzene (**17**) was synthesized differently in this work [52].

#### 3.2.1. General Synthesis Procedure for the Synthesis of Compounds **13–17**

The compound 2-Bromo-4,5-dimethoxybenzenemethanol (**7**) (5 mmol), the corresponding benzene derivatives (**8**–**12**) (5 mmol), and AlCl_3_ (7 mmol) were dissolved in 30 mL of dry CH_2_Cl_2_. The solution was cooled to 0 °C in an ice bath and stirred for 24 h. The reaction mixture was quenched by ice-cold water (20 mL) to remove unreacted AlCl_3_. The organic phase was separated, and the water phase was extracted with CH_2_Cl_2_ (2 × 30 mL). The combined organic layers were dried over anhydrous Na_2_SO_4_, and the solvent was evaporated. Then, the crude products were separated on a silica gel column by using hexane/EtOAc to obtain the pure products.

#### 3.2.2. 1-Bromo-2-(2-bromo-4-methoxybenzyl)-4,5-dimethoxybenzene (**13**)

Yield: 83%, Rf: 0.53, cream solid. M.p. 84–86 °C. **^1^H-NMR (400 MHz, CDCl_3_)** δ: 7.15 (1H, d, *J* = 2.6 Hz, Ar-H), 7.06 (1H, s, Ar-H), 6.89 (1H, d, *J* = 8.5 Hz, Ar-H), 6.77 (1H, dd, *J* = 8.5 Hz, 2.6 Hz, Ar-H), 6.57 (1H, s, Ar-H), 4.06 (2H, s, C-H), 3.87 (3H, s, OCH_3_), 3.78 (3H, s, OCH_3_), 3.75 (3H, s, OCH_3_). **^13^C-NMR (100 MHz, CDCl_3_)** δ: 158.6 (OC), 148.2 (OC), 138.7 (OC), 131.2 (C), 131.2 (C), 130.7 (CH), 124.8 (C-Br), 118.0 (CH), 115.6 (C-Br), 114.7 (CH), 113.6 (CH), 113.5 (CH), 56.2 (OCH_3_), 56.0 (OCH_3_), 55.5 (OCH_3_), 40.6 (CH_2_). IR (cm^−1^, CH_2_Cl_2_): 3080, 3001, 2906, 2837, 1603,1567, 1504, 1463, 1435, 1379, 1341, 1256, 1218, 1162, 1031, 961, 845. Anal. Calcld for C_16_H_16_Br_2_O_3_; C, 46.18; H,3.88. Found: C, 45.88; H, 3.90.

#### 3.2.3. 1-Bromo-4,5-dimethoxy-2-(5-methoxy-2-methylbenzyl)benzene (**14**)

Yield: 86%, Rf: 0.53, white solid. M.p. 93–95 °C. **^1^H NMR (400 MHz, CDCl_3_)** δ: 7.03 (1H, s, Ar-H), 6.99 (1H, dd, *J* = 8.4, 1.6 Hz, Ar-H), 6.78 (1H, s, Ar-H), 6.76 (1H, d, *J* = 8.4 Hz, Ar-H), 6.67 (1H, s, Ar-H), 3.98 (2H, s, C-H), 3.85 (3H, s, OCH_3_), 3.81 (3H, s, OCH_3_), 3.74 (3H, s, OCH_3_), 2.22 (3H, s, CH_3_),**^13^C NMR (100 MHz, CDCl_3_)** δ: 155.2 (OC), 148.3 (OC), 147.8 (OC), 132.2 (C), 130.6 (C), 129.7 (CH), 128.0 (C), 127.7 (CH), 115.4 (C-Br), 114.7 (CH), 113.8 (CH), 110.2 (CH), 56.2 (OCH_3_), 55.9 (OCH_3_), 55.5 (OCH_3_),35.4 (CH_2_), 20.6 (CH_3_). Anal. Calcld for C_17_H_19_Br_2_O_3_; 58.13; H, 5.45. Found: C, 57.65; H, 5.39.

#### 3.2.4. 1-Bromo-2-(4-bromo-2,5-dimethoxybenzyl)-4,5-dimethoxybenzene (**15**)

Yield: 90%, Rf: 0.43, white solid. M.p. 104–106 °C. **^1^H NMR (400 MHz, CDCl_3_)** δ: 7.06 (1H, s, Ar-H), 7.03 (1H, s, Ar-H), 6.67 (1H, s, Ar-H), 6.64 (1H, s, Ar-H), 3.96 (2H, s, C-H), 3.85 (3H, s, OCH_3_), 3.80 (3H, s, OCH_3_), 3.76 (3H, s, OCH_3_), 3.74 (3H, s, OCH_3_), **^13^C NMR (100 MHz, CDCl_3_)** δ: 151.7 (OC), 150.1 (OC), 148.4 (OC), 148.1 (OC), 131.3 (C), 128.6 (C), 115.8 (C-Br), 115.5, 114.7 (CH), 114.6 (CH), 113.6 (CH), 109.2 (CH), 56.9 (OCH_3_), 56.1 (2.OCH_3_), 56.0 (OCH_3_), 35.4 (CH2). **IR (cm^−1^, CH_2_Cl_2_):** 2934, 2838, 1602, 1449, 1463, 1378, 1257, 1212, 1163, 1034, 852. Anal. Calcld for C_17_H_18_Br_2_O_4_: C, 45.77; H, 4.07. Found: C, 45.60; H, 4.05

#### 3.2.5. 1-Bromo-2-(2,5-dimethoxybenzyl)-4,5-dimethoxybenzene (**16**)

Yield: 75%, Rf: 0.56, cream solid. M.p. 97–99 °C. **^1^H NMR (400 MHz, CDCl_3_)** δ: 7.03 (1H, s, Ar-H), 6.81 (1H, d, *J* = 8.0 Hz, Ar-H), 6.72 (1H, dd, *J* = 8.0 Hz, 3.0 Hz, Ar-H), 6.69 (1H, s, Ar-H), 6.57 (1H, d, *J* = 3.0 Hz, Ar-H), 3.99 (2H, s, C-H), 3.86 (3H, s, OCH_3_), 3.81 (3H, s, OCH_3_), 3.75 (3H, s, OCH_3_), 3.71 (3H, s, OCH_3_), **^13^C NMR (100 MHz, CDCl_3_)** δ: 153.5 (OC), 151.6 (OC), 148.4 (OC), 147.9 (OC), 131.7 (C), 129.6 (C), 116.5 (CH), 115.4 (C-Br), 114.7 (CH), 113.7 (CH), 111.2 (CH), 111.1 (CH), 56.1 (OCH_3_), 55.9 (OCH_3_), 55.8 (OCH_3_), 55.6 (OCH_3_), 35.5 (CH_2_). **IR (cm^−1^, CH_2_Cl_2_)**: 2997, 2937, 2834, 1602, 1574, 1500, 1463, 1436, 1379, 1257, 1217, 1163, 1112, 1030, 933, 860. Anal. Calcld for C_17_H_19_BrO_4_; C, 55.60; H,5.21. Found: C, 54.95; H, 5.22.

#### 3.2.6. 1-Bromo-2-(2,5-dimethoxybenzyl)-4,5-dimethoxybenzene (**17**)

1-(2-bromo-4,5dimethoxybenzyl)-2,3,4-trimethoxybenzene (**17**) was synthesized by a different method than that of described previously [52].

Yield: 92%, Rf: 0.30, white solid. M.p. 79–80 °C, Lit. Mp:75–77 °C, **^1^H NMR (400 MHz, CDCl_3_)** δ: 7.02 (1H, s, Ar-H), 6.68 (1H, d, *J* = 8.5 Hz, Ar-H), 6.63 (1H, s, Ar-H), 6.57 (1H, d, *J* = 8.6 Hz, Ar-H), 3.96 (2H, s, C-H), 3.87 (3H, s, OCH_3_), 3.84 (3H, s, OCH_3_), 3.83 (3H, s, OCH_3_), 3.82 (3H, s, OCH_3_), 3.73 (3H, s, OCH_3_). **^13^C NMR (100 MHz, CDCl_3_)** δ: 152.6 (OC), 152.0 (OC), 148.5 (OC), 148.1 (OC), 142.5 (OC), 132.5 (C), 126.1 (C), 124.3 (CH), 115.5 (CH), 114.7 (C-Br), 113.8 (CH), 107.3 (CH), 56.4 (OCH_3_), 56.3 (OCH_3_), 56.2 (OCH_3_), 56.1 (OCH_3_), 56.0 (OCH_3_), 35.4 (CH_2_). Anal. Calcld for C_18_H_21_BrO_5_; C, 54.42; H, 5.33. Found: C, 54.10; H, 5.21.

#### 3.2.7. General Procedure for the Synthesis of Bromophenols **18–21**

Diaryl methane compounds (**13**–**17**) were dissolved in CH_2_Cl_2_. The solutions were cooled to 0 °C. To these solutions, for each methoxy group in the structure of these compounds, 3 equivalents of BBr_3_ were added dropwise under N_2_ atmosphere. Then, the mixtures were stirred at rt for 24 h. The reaction medium was cooled to 0 °C. Ice (20 g) and CH_2_Cl_2_ (50 mL) were added to the reaction medium and the organic phases were separated. Then, the water phase was extracted with ethyl acetate (2 × 50 mL). The organic layers were combined, dried over anhydrous Na_2_SO_4_ and the solvents were evaporated. The residue was crystallized from EtOAc/Hexane.

#### 3.2.8. 4-Bromo-5-(2-bromo-4-hydroxybenzyl)benzene-1,2-diol (**18**)

Yield: 75%, white solid. M.p. 150–153 °C. **^1^H NMR (400 MHz, Acetone-d_6_)** δ: 6.99 (1H, d, *J* = 2.5 Hz, Ar-H), 6.95–6.91 (1H, m, Ar-H), 6.80 (1H, d, *J* = 8.4 Hz, Ar-H), 6.68 (1H, dd, *J* = 8.4, 2.5 Hz, Ar-H), 6.36 (1H, s, Ar-H), 3.80 (2H, s, CH). **^13^C NMR (100 MHz, Acetone-d_6_)** δ: 157.7 (OC), 145.4 (OC), 140.3 (OC), 132.3 (C), 130.8 (C), 127.2 (C-Br), 120.0 (CH), 119.8 (CH), 117.8 (C-Br), 117.2 (CH), 115.7 (CH), 113.3 (CH), 40.2 (CH2). **IR (cm^−1^, CH_2_Cl_2_)**: 3324, 2392, 1686, 1605, 1490, 1429, 1350, 1274, 1226, 1185, 1143, 1031, 919, 872. Anal. Calcld for C_13_H_10_Br_2_O_3_; C41.75; H, 2.69. Found: C, 41.97; H, 2.63.

#### 3.2.9. 4-Bromo-5-(2-bromo-4-hydroxybenzyl)benzene-1,2-diol (**19**)

Yield: 78%, cream solid. M.p.157–159 °C. **^1^H NMR (400 MHz, Acetone-d_6_)** δ 6.91 (1H, s, Ar-H), 6.74 (1H, dd, *J* = 8.0, 2.0 Hz, Ar-H), 6.65 (1H, s, Ar-H), 6.64 (1H, d, *J* = 8.0 Hz, Ar-H), 6.49 (1H, s, Ar-H), 3.75 (2H, s, CH), 2.04 (3H, s, CH_3_). **^13^C NMR (100 MHz, Acetone-d_6_)** δ 153.6 (OC), 145.5 (OC), 145.1 (OC), 132.2 (C), 131.6 (C), 128.5 (C), 127.0 (CH), 122.1 (CH), 119.5 (CH), 118.1 (CH), 115.6 (C-Br), 113.4 (CH), 35.4 (CH_2_), 20.3 (CH_3_). **IR (cm^−1^, CH_2_Cl_2_)**: 3200, 2389, 1502, 1502, 1421, 1356, 1276, 1182, 1045, 919, 814. Anal. Calcld for C_14_H_13_BrO_3_; C 54.39; H, 4.24. Found: C, 54.10; H, 4.06.

#### 3.2.10. 4-Bromo-5-(4-bromo-2,5-dihydroxybenzyl)benzene-1,2-diol (**20**)

Yield: 70%, cream solid. M.p.159–161 °C. **^1^H NMR (400 MHz, Acetone-d_6_)** δ 8.11 (1H, s, OH), 8.08 (1H, s, OH), 8.04 (1H, s, OH), 7.99 (1H, s, OH), 6.91 (1H, s, Ar-H), 6.89 (1H, s, Ar-H), 6.54 (1H, s, Ar-H), 6.39 (1H, s, Ar-H), 3.69 (2H, s, CH). **^13^C NMR (100 MHz, Acetone-d_6_)** δ 149.4 (OC), 147.9 (OC), 145.7 (OC), 145.5 (OC), 131.3 (C), 128.5 (C), 119.7 (CH), 119.4 (CH), 118.3 (C-Br and CH), 113.5 (CH), 107.2 (C-Br), 35.3 (CH_2_). **IR (cm^−1^, CH_2_Cl_2_)**: 3215, 1685, 1602, 1500, 1420, 1355, 1274, 1192, 997, 935, 873. Anal. Calcld for C_13_H_10_Br_2_O_4_; C 40.03; H, 2.58. Found: C, 40.41; H, 2.50.

#### 3.2.11. 4-Bromo-5-(2,5-dihydroxybenzyl)benzene-1,2-diol (**21**)

Yield: 82%, dark yellow. M.p.150–153 °C. **^1^H NMR (400 MHz, Acetone-d_6_)** δ 6.91 (1H, s, Ar-H), 6.58 (1H, d, *J* = 8.5 Hz, Ar-H), 6.53 (1H, s, Ar-H), 6.40 (1H, dd, *J* = 8.5, 3.0 Hz, Ar-H), 6.27 (1H, d, *J* = 3.0 Hz, Ar-H), 3.76 (2H, s, CH). **^13^C NMR (100 MHz, Acetone-d_6_)** δ 151.15 (OC), 148.65 (OC), 145.52 (OC), 145.20 (OC), 145.18 (OC), 132.06 (C), 128.15 (C), 119.61 (CH), 118.31 (CH), 117.44 (CH), 117.44 (CH), 116.28 (C-Br), 114.22 (CH), 35.59 (CH_2_). **IR (cm^−1^, CH_2_Cl_2_)**: 3217, 2395, 2230, 1687, 1595, 1502, 1422, 1279, 1195, 1049, 876.Anal. Calcld for C_13_H_11_BrO_4_; C 50.18; H, 3.56. Found: C, 50.20; H, 3.45.

### 3.3. Biochemical Studies

#### 3.3.1. Enzyme Activity Assays

In this work, the in vitro inhibition effects of bromophenols (**13**–**21**) on AChE activity were determined by Ellman’s method [76], as previously described [77]. The results were recorded spectrophotometrically at 412 nm. Acetylthiocholine iodide (AChI) was used as substrate, according to a prior study [78]. Both hCA isoforms were purified by using the Sepharose-4B-L-Tyrosine-sulfanilamide affinity technique [79]. Then, the purity of these CA isoenzymes was defined via the SDS-PAGE purity technique [80,81]. Furthermore, the hCA activity was determined using the esterase method at 348 nm, according to the method of Verpoorte et al. [82] and as given in prior studies [83,84].

#### 3.3.2. Enzyme Inhibition Assays

In order to investigate the in vitro inhibitory mechanisms of bromophenols (**13**–**21**), kinetic studies were performed at different concentrations of bromophenols and various substrates [85,86]. From the Lineweaver–Burk graphs, the IC_50_ and K_i_ values of bromophenol (**13**–**21**) derivatives were calculated, and the inhibition type of bromophenols (**13**–**21**) against AChE and hCAs was determined, as given in prior studies [87,88,89]. 

#### 3.3.3. Statistical Analyses

Statistical analyses were performed via an unpaired Student’s *t*-test with the use of the statistical program IBM SPSS Statistics 20. The results were recorded as means with their standard deviation (SD). *p* < 0.05 was the minimum significance level.

## 4. Conclusions

In the current study, new bromophenols were synthesized, and their hCAs and AChE inhibitory properties were investigated. The presence of different biologically functional groups (-OH, -OCH3, and -Br) in aromatic scaffolds of synthesized compounds influenced the activity of the studied enzymes. Our findings indicate that the investigated compounds **13**–**21** exhibited efficient hCA I, II, and AChE inhibition effects in the low nanomolar levels. These experimental findings confirm that substituted methoxy (-OCH3) and bromophenols may be used as leads for generating potent CAI and AChE inhibitors associated with some global disorders, including AD, epilepsy, and glaucoma.

## Data Availability

Data are provided in a publicly accessible repository.

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
