# Peer review of "Synthesis of Novel Bromophenol with Diaryl Methanes—Determination of Their Inhibition Effects on Carbonic Anhydrase and Acetylcholinesterase"

_molecules, 2022, doi:10.3390/molecules27217426_

Round 1
Reviewer 1 Report
This article is covering synthesis of novel bromophenols and their inhibition effects on carbonic anhydrase and acetylcholinesterase.
The design of synthetic strategies of this class of analogs of natural bromophenols and verification of their inhibitory activities is of utmost importance. The synthetic approaches to these target compounds are complex and very laborious. Authors proposed the alternative techniques by relatively simple approach and test them as potential inhibitors of carbonic anhydrase. This will definitely constitute the important goals and noveltyof this important paper!
The following suggested minor changes and recommendations should be introduced before the publication of the manuscript.
1. Page 1, line 25, replace “researches” with “research”
2. Page 1, line 31, 34, numbers for compounds 2 and 4 should be in bold.
3. Page 1, line 43, replace “very well know” with “concluded”
4. Page 2, line 52, remove “class”
5. Page4, figure 2, each plot should be identified as one of three types of inhibition: competitive, non-competitive or mixed. The same information should be inserted into the text of the manuscript referring to Ki ranges.
6. Page 8, line 247, line 327. The sentence “The crystallization of the residue with EtOAc/Hexane” should be corrected to “ The residue was crystalized from EtOAc/Hexane”
7. Page 9, line 304, numbers for compounds 13-21 should be in bold!
The manuscript is of good quality and importance and is written and edited in order to meet the standard for the articles published in Molecules.Thus, I certainly recommend it for publication after the correction of these suggested minor changes.
Author Response
RESPONSES TO REVIEWER-1
This article is covering synthesis of novel bromophenols and their inhibition effects on carbonic anhydrase and acetylcholinesterase.
The design of synthetic strategies of this class of analogs of natural bromophenols and verification of their inhibitory activities is of utmost importance. The synthetic approaches to these target compounds are complex and very laborious. Authors proposed the alternative techniques by relatively simple approach and test them as potential inhibitors of carbonic anhydrase. This will definitely constitute the important goals and novelty of this important paper!
RESPONSE: Many thanks to the reviewer due to his/her positive opinion.
The following suggested minor changes and recommendations should be introduced before the publication of the manuscript.
RESPONSE: All suggested corrections were made by authors.
- Page 1, line 25, replace “researches” with “research”
RESPONSE: “researches” replaced with “research”.
- Page 1, line 31, 34, numbers for compounds 2 and 4 should be in bold.
RESPONSE: The numbers for compounds 2 and 4 were given in bold.
- Page 1, line 43, replace “very well know” with “concluded”
RESPONSE: “very well know” was replaced with “concluded”.
- Page 2, line 52, remove “class”
RESPONSE: “class” vas removed.
- Page4, figure 2, each plot should be identified as one of three types of inhibition: competitive, non-competitive or mixed. The same information should be inserted into the text of the manuscript referring to Ki ranges.
RESPONSE: Also, all the novel synthesized a series of bromophenols (13-21) demonstrated competitive inhibition against hCA I and II isozymes and AChE enzyme. This is stated in the text for each enzyme used in the study.
- Page 8, line 247, The sentence “The crystallization of the residue with EtOAc/Hexane” should be corrected to “The residue was crystalized from EtOAc/Hexane”
RESPONSE: “The crystallization of the residue with EtOAc/Hexane” was corrected as “The residue was crystalized from EtOAc/Hexane”.
- Page 9, line 304, numbers for compounds 13-21 should be in bold!
RESPONSE: The numbers for compounds 13-21 were given in bold.
The manuscript is of good quality and importance and is written and edited in order to meet the standard for the articles published in Molecules. Thus, I certainly recommend it for publication after the correction of these suggested minor changes.
RESPONSE: We tanks to reviewer due to his/her positive opinion.
Reviewer 2 Report
The manuscript submitted by Oztaskin, Goksu, Gulcin and coll. reports the synthesis and enzyme inhibition studies of brominated benzylbenzene-diol.
Corrections/comments must be made for a hypothetical publication:
- Abstract: no future work in this part
- Introduction: instead of bromophenol X or compound X, give IUPAC names of molecules 1-6
- To complete this study the compound 4-(2-bromo-4,5-dihydroxybenzyl)benzene-1,2,3-triol (from 17 with BBr3) must be synthesized and tested
- Scheme 1: compounds numbers in bold
- 2.1 chemistry: Are the second isomers of the products observed? A comment on this regioselectivity of the alkylation reaction should be added
- 2.2. Biochemistry: the authors highlight the positive effect of certain chemical functions on certain positions, a very positive point, but no explanation or even hypothesis on the biological mechanism of action is reported. This point needs to be corrected.
- 3.1. General: 1H and 13C with superscript numbers
- 3.2.1. (2-bromo-4,5-dimethoxyphenyl)methanol (7): this paragraph is not necessary and should be deleted
- 3.2.2. General synthesis procedure for the synthesis of compounds 13-17: For the silica gel columns composition of hexane/EtOAc eluent and Rf of compounds 13-17 must be added. Why hexane, it is toxic solvent especially when enzyme inhibition tests were carried out.
- 3.2.11. 4-bromo-5-(4-bromo-2,5-dihydroxybenzyl)benzene-1,2-diol (20): 118.3(C-Br, CH)? Experimental part must be carefully checked.
- Supplementary materials containing all NMR spectra , IR spectra…. must completed the manuscript
The work is not enough detailed; I do not recommend publication of the manuscript in Molecules.
Author Response
RESPONSES TO REVIEWER-2
The manuscript submitted by Oztaskin, Goksu, Gulcin and coll. reports the synthesis and enzyme inhibition studies of brominated benzylbenzene-diol.
Corrections/comments must be made for a hypothetical publication:
RESPONSE: All corrections suggested by reviewer 2 were performed by us.
- Abstract: no future work in this part
RESPONSE: The indicated sentence was corrected as “The results show that they can be used for the treatment of glaucoma, epilepsy, Parkinson’s and Alzheimer’s disease (AD) after some imperative pharmacological studies to reveal drug potential.”.
- Introduction: instead of bromophenol X or compound X, give IUPAC names of molecules 1-6
RESPONSE: It was corrected as advised.
- To complete this study the compound 4-(2-bromo-4,5-dihydroxybenzyl)benzene-1,2,3-triol (from 17 with BBr3) must be synthesized and tested
RESPONSE: It takes a long time to purchase chemicals from Europe or USA. We did not have 1,2,3-trimethoxybenzene. However, we were able to find 0.5 g from our friends and at the request of the referee, we tried to synthesize the mentioned compound from compound 17. However, we failed to synthesize 4-(2-bromo-4,5-dihydroxybenzyl)benzene-1,2,3-triol by demethylation of compound 17 with BBr3. If the referee insists on the synthesis of this compound, we will order 1,2,3-trimethoxybenzene and resynthesize the mentioned compound. It takes at least 6 months to carry out this extra experiment, together with the supply of the necessary chemicals.
- Scheme 1: compounds numbers in bold
RESPONSE: The compound numbers were given as bold.
- 2.1 chemistry: Are the second isomers of the products observed? A comment on this regioselectivity of the alkylation reaction should be added.
RESPONSE: As we know from our early studies, the second isomers of the product 13 was not observed. The second isomer of compound 13, 14 and 16 may be occurring, however, we did not separate. Because, as can be understood from the yields of the products obtained, the formation of the second isomers may have occurred with low yields. Due to this possibility, regioselectivity was not mentioned in the article. As can be seen from previous studies, compounds 15 and 17 are formed as a single product.
- 2.2. Biochemistry: the authors highlight the positive effect of certain chemical functions on certain positions, a very positive point, but no explanation or even hypothesis on the biological mechanism of action is reported. This point needs to be corrected.
RESPONSE: The proposed interaction between the most powerful bromophenols (20) and CA II isoforms was illustrated in Figure 3. Also, the following information was given for this purpose: “The proposed interaction between the most powerful bromophenols (20) and CA II isoforms was illustrated in Figure 3. Bromophenol (20) has two dihydroxy benzyl rings. A second hydrogen bond was modeled between the oxygen atom, which attached to the -OH group at the phenol moiety in the ortho- position and the amide NH of Thr199, a universally conserved amino acid residue in CAs. Thus, of phenolic compounds and derivatives bind non-classically to CA, providing clues for the identification of new types of CA inhibitors. Such inhibition mechanisms of phenolic compounds including bromophenols are known [68,69].”.
- 3.1. General: 1H and 13C with superscript numbers
RESPONSE: 1H and 13C were given with superscript numbers.
- 3.2.1. (2-bromo-4,5-dimethoxyphenyl)methanol (7): this paragraph is not necessary and should be deleted
RESPONSE: The section of 3.2.1. was removed.
- 3.2.2. General synthesis procedure for the synthesis of compounds 13-17: For the silica gel columns composition of hexane/EtOAc eluent and Rf of compounds 13-17 must be added. Why hexane, it is toxic solvent especially when enzyme inhibition tests were carried out.
RESPONSE: We used it for separations because we have hexane. Similarly, hexane was used in the purifications in our previous studies. Since these substances were completely dried, they had no effect on enzyme studies. The Rf of compounds were added as advised.
- 3.2.11. 4-bromo-5-(4-bromo-2,5-dihydroxybenzyl)benzene-1,2-diol (20): 118.3(C-Br, CH)? Experimental part must be carefully checked.
RESPONSE: The original 13C-NMR spectra was checked carefully and the advised data were corrected.
- Supplementary materials containing all NMR spectra, IR spectra…. must completed the manuscript
RESPONSE: Supplementary materials containing all NMR spectra, IR spectra was uploaded.
The work is not enough detailed; I do not recommend publication of the manuscript in Molecules.
RESPONSE: The corrections given attached file were performed in the revised manuscript, we believe that after this revision, the referee will change this negative opinion and act positively.
Round 2
Reviewer 2 Report
With supplementary materials, the manuscript could be published